# Gut Microbiome of a Multiethnic Community Possessed No Predominant Microbiota

**DOI:** 10.3390/microorganisms9040702

**Published:** 2021-03-29

**Authors:** Wei Wei Thwe Khine, Anna Hui Ting Teo, Lucas Wee Wei Loong, Jarett Jun Hao Tan, Clarabelle Geok Hui Ang, Winnie Ng, Chuen Neng Lee, Congju Zhu, Quek Choon Lau, Yuan-Kun Lee

**Affiliations:** 1Department of Microbiology & Immunology, Yong Loo Lin School of Medicine, National University of Singapore, 5 Science Drive 2, Singapore 117545, Singapore; micwwtk@nus.edu.sg (W.W.T.K.); anna_tht@hotmail.com (A.H.T.T.); 2Functional Food Forum, Faculty of Medicine, University of Turku, 20014 Turku, Finland; 3School of Life Sciences & Chemical Technology, Ngee Ann Polytechnic, 535, Clementi Road, Singapore 599489, Singapore; lucas_wwl@hotmail.com (L.W.W.L.); jarett35@gmail.com (J.J.H.T.); clarabelleagh@outlook.com (C.G.H.A.); winnieng2201@gmail.com (W.N.); john_zhu@np.edu.sg (C.Z.); lau_quek_choon@np.edu.sg (Q.C.L.); 4Department of Surgery, National University of Hospital, Tower Block, 1E Kent Ridge Road, Singapore 119228, Singapore; surlcn@nus.edu.sg

**Keywords:** cytokines, faecal microbiome, dietary habits, probiotics, immune response, multicultural dietary habit

## Abstract

With increasing globalisation, various diets from around the world are readily available in global cities. This study aimed to verify if multiethnic dietary habits destabilised the gut microbiome in response to frequent changes, leading to readily colonisation of exogenous microbes. This may have health implications. We profiled Singapore young adults of different ethnicities for dietary habits, faecal type, gut microbiome and cytokine levels. Subjects were challenged with *Lactobacillus casei*, and corresponding changes in microbiome and cytokines were evaluated. Here, we found that the majority of young adults had normal stool types (73% Bristol Scale Types 3 and 4) and faecal microbiome categorised into three clusters, irrespective of race and gender. Cluster 1 was dominated by *Bacteroides*, Cluster 2 by *Prevotella,* while Cluster 3 showed a marginal increase in *Blautia*, *Ruminococaceae* and *Ruminococcus*, without a predominant microbiota. These youngsters in the three faecal microbiome clusters preferred Western high sugary beverages, Southeast Asian plant-rich diet and Asian/Western diets in rotation, respectively. Multiethnic dietary habits (Cluster 3) led to a gut microbiome without predominant microbiota yet demonstrated colonisation resistance to *Lactobacillus*. Although *Bacteroides* and *Prevotella* are reported to be health-promoting but also risk factors for some illnesses, Singapore-style dietary rotation habits may alleviate *Bacteroides* and *Prevotella* associated ill effects. Different immunological outcome was observed during consumption of the lactobacilli among the three microbiome clusters.

## 1. Introduction

Interactions between gut microbes and the human host are expected because of their lifelong association and proximity. Indeed, gut microbiome has been reported to play vital roles in the physiological functions and wellbeing of people [1,2]. *Bifidobacterium* and *Bacteroides* have been aligned with the maturation of host immunity in earlier life [3,4] correction of GI disorders associated with colitis [5], as well as behavioural and physiological abnormalities associated with neurodevelopment disorders [6]. *Prevotella* was reported as positively interfering in energy homeostasis and glucose control [7,8]. The first wave of gut microbes arrives mostly through vertical transmission from mother to child [9]. Upon introduction of solid foods, dietary habit has been demonstrated as the major determining factor in gut microbiome composition, in studies comparing the microbiome of people across geographical regions and of different ethnicities [10,11,12,13]. Overall, high meat protein and fat, high sugar, non-resistant starch Western and Eastern Asian diets are associated with a *Bacteroides–Bifidobacterium* dominated gut microbiome [10,11,12,13], whereas the plant-, fiber- and carbohydrate- (high in resistant starch) rich diet of Southeast Asian and African type favours *Prevotella* in the gut microbiome [10,11,12,13]. These associations may be due to the provision of metabolisable nutrients and the resulting microenvironment. Establishment of the respective stable microbiome compositions facilitates colonisation resistance, in protecting the gut from being colonised by undesirable enteric pathogens arriving with foods [14].

Today, foods around the world are readily available in global cities, such as Singapore, where people consume foods of different origins in random rotation. It was the aim of this study to verify if Singapore’s multiethnic dietary habit would destabilised gut microbiome in response to frequent changes, which may allow readily colonisation of exogenous microbes introduced orally. This may have health implications. In this study, the faecal microbiome and dietary habits of young Singaporeans were profiled and their associations were evaluated. The responses of each diet-determined microbiome cluster to an invading microbe (oral consumption of *Lactobacillus*) were assessed as an intervention model to study the gut microbiota structural stability and gut immune response to the invading microbe. Lactobacilli are bacteria with generally regarded as safe (GRAS) status and have been demonstrated to modulate the gut microbiome profile and immune functions as probiotics.

## 2. Materials and Methods

### 2.1. Study Design

A longitudinal randomised study was performed on 75 healthy young adults with an average age of 19.2 ± 1.9 years (18–30 years) of both genders (male: *n* = 30, female: *n* = 45), and ethnicities (Chinese: *n* = 61, Malay: *n* = 8, Indian: *n* = 2, Caucasian: *n* = 1, Filipino: *n* = 1, Korean: *n* = 1, Vietnamese: *n* = 1). Their BMI were 21.4 ± 2.9 at the baseline of the study. The subjects were recruited from a tertiary institution to minimise age, body weight and other confounding factors. They were instructed to maintain their dietary habit and lifestyle during the study. The total 42-days study included 14 days washout period (baseline; timepoint 1), 14 days ingestion period (two time points of 7 days apart each) and 14 days follow-up period (timepoint 4). At baseline and follow-up periods, the subjects abstained from the consumption of probiotic products. During the ingestion period, 100 mL of fermented milk containing the *Lactobacillus casei* (1 × 10^10^ CFU/mL) was provided every morning for 14 days. The dosage and period of consumption were chosen with reference to the Probiotics Fact Sheet for Health Professionals of the National Institutes of Health Office of Dietary Supplements 2020. The *Lactobacillus casei* was isolated from a local fermented food, identified by API 50 CHL kit (Biomerieux API), and thus is generally regarded as safe (Joint FAO/WHO Expert Committee on Food Additives). A *Lactobacillus* was chosen in this study for lactobacilli have been widely reported to modulate gut microbiome and host physiology (immunity). Milk was chosen as the culture and delivery medium, for dairy products were widely consumed among the subjects (consumed 2–12 times a week). Faecal samples were collected a total of four times at the end of each period. Foods frequency survey was collected at the baseline. Stool characteristics, frequency of defecation, water consumption and medications were recorded daily during the study. The protocol of the study is summarised in the following flow chart (Figure 1).

Inclusion criteria were 18–30 years old healthy adults who had normal BMI, no history of gastrointestinal disorder, not on long-term medication, no planning for overseas trip, able to drink the fermented milk with *Lactobacillus casei* everyday during the study period, able to abstain from any other fermented food products during the study period, and able to sign the informed written consent form. Exclusion criteria in the study were if the participants have allergy or intolerance to a special diet, using antibiotics, antimycotics, antidiarrheal or laxative medication in the 30 days prior to the study, lack of compliance with the study protocol, or participating in any other studies within two weeks prior to the study.

### 2.2. Food Frequency Questionnaire

All the subjects completed an in-house food frequency questionnaire (FFQ) (Appendix A) at baseline. Only 35 FFQ answers were able to be analyzed in this study due to the qualities of responses and missing data. The food items focused on are carbohydrate-rich foods (rice, noodles, cereals, bread, burger, French fries), protein-rich foods (meats, soy protein products), vegetables, fruits, nuts (peanuts, almonds, cashew, walnuts, macadamia, chestnuts), and beverages (coffee, tea, soft drinks, juices). The frequencies of food items that belonged to the same food category were summed up and the total frequency of food items was calculated per week.

### 2.3. Faecal Sample Collection and DNA Extraction

Approximately 1 g of faeces was collected and preserved in a collection tube containing 2 mL of RNAlater^®®^ (Ambion Inc., Austin, TX, USA). A total of 75 faecal samples were collected. After weighing the samples, the faecal homogenate was diluted 10 times with RNAlater and 0.2 mL was washed with phosphate-buffered saline (PBS) (Axil Scientific Pte Ltd., Singapore, Singapore) which was later treated with Tris-SDS and TE-saturated phenol (Sigma-Aldrich, Cor., St. Louis, MO, USA) solution. After vigorously shaking, the supernatant was mixed with phenol/chloroform/isoamyl alcohol (25:24:1) (Sigma-Aldrich, Cor., St. Louis, MO, USA) followed by homogenisation. Sodium acetate and isopropanol precipitated DNA was washed with 70% ethanol. Once dried, the pellet was eluted in TE buffer.

### 2.4. 16s rRNA DNA Sequencing

After DNA was quantified and calculated for polymerase chain reaction (PCR), the normalized 12.5 ng DNA was amplified with KAPA HiFi^TM^ HotStart ReadyMix kit (Roche life science, Inc., Wilmington, MA, USA) at v3 and v4 regions of the 16s rRNA gene. Amplified PCR products were purified with Agencourt^®®^ AMPure XP beads (Beckman Coulter, Inc., Fullerton, CA, USA). The amplicons were added Nextera XT indices and adapter sequences. After two rounds of PCR, the DNA was purified again with the beads and eluted in Tris buffer. The library was quantified with Quanti-iT^TM^ PicoGreen^®®^ dsDNA kit (Invitrogen, Inc., Carlsbad, FA, USA) and qualified with Agilent high sensitivity DNA kit (Agilent Technologies, Inc., Santa Clara, CA, USA) in Agilent 2100 bioanalyzer (Agilent Technologies, Inc., Santa Clara, CA, USA). All the libraries were pooled and quantified with KAPA library quantification kit (Roche life science, Inc., Wilmington, MA, USA) in the ABI 7500 real-time PCR system (Thermo Fisher Scientific, Inc., Waltham, MA, USA). The pooled library was denatured and diluted with NaOH and hybridisation buffer (HT1) (Illumina, Inc., San Diego, CA, USA) until the required titrated concentration. The denatured amplicon library was spiked with the PhiX control library (Illumina, Inc., San Diego, CA, USA) and run in Miseq sequencer (Illumina, Inc., San Diego, CA, USA).

### 2.5. Bioinformatics Analysis and Clustering

Quantitative insights into microbial ecology (QIIME) tool version 1.9.1 [15] was used for 16 s rRNA DNA sequence data. The selected paired reads were filtered and resulted in chimeric sequences by USEARCH v6.1 [16]. OTUs were picked by the open reference method and matched with 97% similarity sequences at Greengenes v13_8 database. Summarised taxa of relative abundance of OTUs revealed the bacterial genera in each sample. Alpha diversity indices were calculated using the OTU table referenced by the same database tree. Chao 1 and Shannon indices for alpha diversity were presented in this study. Unweighted and weighted UniFrac distance matrices for beta diversity were analysed using R package [17].

According to Entero-typing methods [18,19], the Jensen-Shannon distance (JSD) matrix was calculated using the relative abundance of bacterial genera data of Timepoint 2 in R v4. Clustering was performed by the partitioning around medoids (PAM) algorithm. The optimal number of clusters was estimated by Calinski-Harabasz (CH) index and validated by the individual and average silhouette coefficient (Si). Three classified clusters were named Cluster 1 (*n* = 31), Cluster 2 (*n* = 14) and Cluster 3 (*n* = 30). The assigned cluster to individual samples of Timepoint 2 was applied for further visualisation and categorisation of remaining samples.

A constrained Redundancy Analysis (RDA) based on the square root of Bray-Curtis distances (db-RDA) was performed using the relative abundance of genera data by the Canoco5 software package (Microcomputer Power Co, Ithaca, NY, USA). The principal coordinate analysis (PCoA) plots using unweighted and weighted UniFrac distances were visualised for beta diversity analysis by R v4.

### 2.6. Faecal Water Cytokines Analysis

One volume of faecal homogenate was treated with two volumes of 0.01 M Phenyl-methyl-sulfonyl fluoride (PMSF) (Sigma-Aldrich, St. Louis, MO, USA), 1% Bovine Serum Albumin (BSA) and PBS solution. 10 µL of supernatant was diluted 1:3 ratio in assay diluent 1 of the LUNARIS^TM^ Human 11-Plex cytokine kit (AYOXXA Biosystems, Cologne, Germany) and loaded onto a LUNARIS^TM^ BioChip with its standards and blanks. The remaining procedures were followed as per the manufacturer’s guide. Fluorescence from each well of BioChip was read by a fluorescence microscope (Zeiss Axio Imager M2, Carl Zeiss Microscopy, Oberkochen, Germany) and quantified using the LUNARIS^TM^ analysis. 11 cytokines, namely, interleukins (IL) 1-β, -2, -4, -5, -6, -8, -10, -12, granulocyte-macrophage colony-stimulating factor (GM-CSF), interferon gamma (IFNɣ) and tumour necrosis factor alpha (TNFα) were identified for 72 faecal water samples (Cluster 1: *n* = 30; Cluster 2: *n* = 14; Cluster 3: *n* = 28) for four time points. One sample was unable to analyse at Timepoint 2 of Cluster 3 due to the insufficient amount. The concentration of each sample in pg/mL was calculated and converted to pg in g of wet weight of faeces in the collection tube by multiplied the dilution factor. The lower limit of quantification (LLOQ) was used as a cut-off.

### 2.7. Statistical Analysis

To compare the differences of the bacterial community in three clusters, ethnicities, genders and Bristol stool scales shown in the db-RDA plot, a permutational multivariate analysis of variance (PERMANOVA) was performed using the pairwiseAdonis R package [20]. Adjusted *p* values were derived from the post hoc Bonferroni multiple pairwise test and permutation test at 4999. All the data were checked for normality and subsequently analysed by the appropriate statistical methods. Individual major bacterial genera abundances (>1% of total OTU) and frequency of food items were analysed using a non-parametric Mann Whitney U test to compare the different clusters. To identify the correlations between abundances of the major bacteria and total frequency of food items, the non-parametric Spearman correlation test was performed using 35 same samples of bacteria and FFQ data (Cluster 1: *n* = 9; Cluster 2: *n* = 10; Cluster 3: *n* = 16). One-way analysis of variance (ANOVA) and Bonferroni multiple comparisons tests were applied for alpha diversity analysis to compare the clusters and time points. The comparison of bacteria abundances for four time points in each cluster was done by non-parametric matched test of Friedman and Nemenyi post hoc multiple pairwise comparison tests. The mixed-effects model or two-way repeated measures ANOVA followed by Bonferroni multiple comparisons tests were performed for the comparison of cytokines in different time points. All the statistical analyses were performed using GraphPad Prism 8 (GraphPad Software Inc., San Diego, CA, USA).

## 3. Results

### 3.1. Clustering of Basal GI Microbiome

The faecal microbiome of Singapore young adults was segregated into three clusters (Figure 2A, Appendix A). Cluster 1 was dominated by *Bacteroides,* which constitute 30% of the total operational taxonomical units (OTU), whereas *Bacteroides* in Cluster 2 and 3 constituted 8% and 12%, respectively, significantly lower than Cluster 1 (Appendix A). The abundance of *Bacteroides* in Cluster 1 was traded off by a reduced fraction of *Ruminococaceae* and *Prevotella*, as compared to Clusters 2 and 3. Cluster 2 was dominated by *Prevotella* (21% of total OTUs). Whereas *Prevotella* in Cluster 1 and 3 constituted only <1% and 1% of total OTUs, *Bacteroides* and *Ruminococcus* were the lowest in Cluster 2 among the three clusters. Cluster 3 does not have a predominant microbiota, but with statistically higher proportion of *Blautia* (16% vs. Cluster 1, 12%, and Cluster 2, 12%), *Ruminococaceae* (8% vs. Cluster 1, 3%, and Cluster 2, 6%) and *Ruminococcus* (4% vs. Cluster 1, 1%, and Cluster 2, 1%). This showed a comparatively even distribution of major microbiota in Cluster 3.

### 3.2. Dietary Composition and Habit

In general, subjects in the three clusters consumed significantly different food types (Figure 2B), mainly sugary beverages, carbohydrate-rich and protein-rich foods, fruits, nuts, and vegetables (Figure 2B, Appendix A). Cluster 1 consumed more sugary beverages (16.11 ± 6.82 per week), carbohydrate-rich (30.72 ± 12.19 per week) and protein-rich (35.94 ± 13.68 per week) foods mostly in the form of Western fast foods, with a lower frequency of vegetables (5.22 ± 3.03 per week) as compared to Cluster 2 (carbohydrate 18.38 ± 7.60, protein 23.96 ± 14.53, vegetables 11.00 ± 4.57 per week).

The diet of Cluster 2 was typically of the plant-rich Southeast Asian type (vegetables), and Cluster 3 adopted the typical dietary habit of Singaporeans in interchanging Western and Asian foods. Overall, cluster 2 ate the lowest frequency of most food types (1 out of 6) among all clusters. However, it consumed more vegetables (11.00 ± 4.57 per week) as compared to Cluster 1 (vegetables 5.22 ± 3.03 per week) and Cluster 3 (vegetables 5.53 ± 2.57 per week). As indicated in Appendix A, Cluster 3 consumed the highest frequency of most food types (4 out of 6). Carbohydrate-rich and protein-rich foods, fruits and nuts were the most consumed foods in cluster 3 among all clusters, but the intermediate frequency of sugary beverages and vegetables compared to the other two clusters (Appendix A). Thus, Cluster 3 is located in a different plane and between Clusters 1 and 2 in the square root of Bray Curtis distance-based Redundancy Analysis (db-RDA) (Figure 2B), although the adjusted *p*-value was not significantly different from Cluster 1 (Appendix A).

### 3.3. Correlation between Diet and Microbiota

In the heatmap correlation between major faecal bacteria and weekly frequency of food type consumption (Figure 2C), *Bacteroides* was found to correlate positively although not statistically significantly with carbohydrate-rich and protein-rich foods and sugary beverages. *Prevotella* was found to be positively and significantly correlated with vegetables, and negatively correlated with carbohydrate-rich, protein-rich foods, fruit, nuts, and sugary beverages, whereas *Blautia* positively correlated with carbohydrate-rich, protein-rich foods, fruits, nuts and sugary beverages; *Ruminococaceae* positively correlated with carbohydrate-rich, protein-rich foods and vegetables; and *Ruminococcus* positively correlated with carbohydrate-rich, protein-rich foods, fruits and nuts. Other microbiota was positively or negatively correlated with the various food types.

### 3.4. Effects of Ethnicities, Genders and Types of Bristol Stool Scale in Association with GI Microbiome

As shown in Appendix A, the major ethnic group in this study was Chinese (*n* = 61), whose faecal microbiome at baseline could be differentiated into three clusters. Malays (*n* = 8) Indians (*n* = 2) were randomly distributed among the three clusters. This might suggest that ethnicity was not a determining factor in the clustering of faecal microbiome among these Singapore youngsters. The rest of the ethnicities, namely Caucasian, Filipino, Korean and Vietnamese, comprised one subject each, and thus could not be accounted for in evaluating the effect of ethnicity.

The effect of gender on faecal microbiome is shown in Appendix A where the same gender was found to be significantly differentiated between different clusters at baseline and at all time points. However, both genders intermingled within the same cluster and thus clustering of faecal microbiota at baseline was not influenced by gender.

Among the seven Bristol stool scale types [21], type 3 and 4 stools are normal in stool shape and consistency, but type 1 is constipated and type 6 diarrhoetic stool. Generally, the microbiome in healthy stool (type 4) could be differentiated into clusters according to dietary habits (Appendix A). In subjects with type 4 stool, microbiome from Cluster 1 was significantly different from that of Clusters 2 and 3 at baseline (Timepoint 1) and at Timepoint 3 during *Lactobacillus* consumption, and also significantly different from Cluster 3 at Timepoint 4 after the cessation of consumption (Appendix A). In stool type other than 4 (types 1, 2, 3, 5 and 6), the differences in microbiome distribution were not statistically obvious (Appendix A), as the subject numbers were few.

### 3.5. Biodiversity of GI Microbiome

Chao 1′s and Shannon’s indices estimated that the species richness and evenness between the clusters at baseline (Figure 3A,B) and between the time points in each cluster (Appendix A) were comparable statistically (Appendix A), apart from, Timepoint 3 in Cluster 1, where there were more species than Timepoint 2 in the same cluster (adjusted *p*= 0.014) (Appendix A).

Both unweighted (Figure 3C) and weighted (Figure 3D) UniFrac distances were significantly different between the clusters at baseline (Appendix A) showing that the types and quantities of species were different between clusters at baseline. In Cluster 1, there was significant difference among all time points but no difference between Timepoint 3 and 4 in both unweighted and weighted UniFrac distances (Appendix A). In Cluster 2, each time point varied from the others in both unweighted and weighted UniFrac distances (Appendix A). Unweighted and weighted UniFrac distances at all time points were different from each other in Cluster 3, but not between Timepoint 1 and 2 (Appendix A).

### 3.6. Introduction of Exogenous Lactobacillus: Cluster 1

Most of the faecal major microbiome profile of Cluster 1 did not appear to alter significantly during the *Lactobacillus* administration (Figure 4A). On examination of the respective microbiota which constitutes >1% of total OTUs, only *Lactobacillus* showed a consistent trend (*p* < 0.05) of increase in abundance during the time of *Lactobacillus* consumption (Timepoints 2 and 3) (Appendix A). Its abundance returned to the baseline (Timepoint 1) on cessation of consumption (Timepoint 4) (Figure 4A).

As shown in Figure 4B and Appendix A, faecal water cytokines IL-1β, -2, -8, -12 and TNFα were detectable, while IL-4, -5, -6, -10, IFNɣ, and GM-CSF were near to detection level across all time points in Cluster 1. Both cytokine IL-1β and -2 increased but IL-8, -12 and TNFα decreased at Timepoint 2. At Timepoint 3, IL-1β dropped nearly back to the level of Timepoint 1 but IL-2, -8, -12 and TNFα increased. Apart from IL-8, IL-1β, -2, -12 and TNFα increased among detectable cytokines at Timepoint 4. There was no significant difference in cytokine levels between genders at the same time points (Appendix A).

### 3.7. Introduction of Exogenous Lactobacillus: Cluster 2

The faecal microbiome in Cluster 2 showed a significant reduction in the abundance of *Prevotella* at Timepoint 4 but not during *Lactobacillus* administration (Figure 5A and Appendix A). All other major faecal microbiota and *Lactobacillus* did not show significant variation at Timepoints 2, 3 and 4, as compared to their basal levels (Timepoint 1).

The basal (Timepoint 1) levels of faecal water cytokine IL-2, -12 and TNFα in Cluster 2 (Figure 5B) were 3–6 times higher than the basal levels in Cluster 1 (Appendix A). This implies higher lymphocytic and pro-inflammatory activities. Upon consumption of the lactic acid bacterium, the level of the pro-inflammatory IL-1β and -8 (Timepoint 3) and TNFα (Timepoint 2) increased, but the level of regulatory cytokine IL-2, -12 and TNFα (Timepoint 3) decreased. After cessation of *Lactobacillus* consumption, IL-8 decreased by 3.8 times but, IL-1β increased by 7.2 times. These showed that the *Lactobacillus* had a stronger immune-modulating effect on subjects in Cluster 2 than Cluster 1. No gender differences were found in cytokine levels at the same time point (Appendix A).

### 3.8. Introduction of Exogenous Lactobacillus: Cluster 3

In Cluster 3, upon consumption of the *Lactobacillus*, some of the faecal microbiota profile was attenuated at Timepoint 4, in comparison with Timepoints 1, 2 and 3 (Figure 6A). The differentiation was due to a significant reduction in the abundance of *Bifidobacterium*, *Collinsella* and *Phascolarctobacterium* at Timepoint 4 (Appendix A). There was no difference in the abundances of other major microbiota, namely *Blautia, Prevotella* and *Bacteroides* over all time points. *Lactobacillus* abundance increased during consumption of the *Lactobacillus* (Timepoints 2 and 3) and returned to the basal level at Timepoint 4.

As shown in Figure 6B and Appendix A, the cytokine IL-8 decreased progressively from Timepoint 1 through Timepoint 4, whereas the level of IL- 2, -12 and TNFα decreased at Timepoint 2, but returned to the basal level at Timepoint 4. The level of IL-1β measured was variable across the time points, decreased at Timepoint 2 as compared to Timepoint 1, increased at Timepoint 3 and decreased again at Timepoint 4. The levels of cytokines were comparable between genders at the same time points (Appendix A).

## 4. Discussion

The subjects recruited in the study were young adults mostly of the same race (Chinese: 61/75 = 81%), falling within a narrow range of age and basal metabolic rate (BMI). Moreover, subjects of different ethnicities (Malay and Indian) were found within the same microbiome clusters as those of the Chinese. Therefore, possible confounding factors, such as ethnicity, age and obesity, could be eliminated from the interpretation of the biodata and their association. It should be highlighted that *Prevotella* dominated Cluster 2 subjects were not heavier than *Bacteroides* Cluster 1, although BMI has been reported as negatively associated with *Bacteroides* [22]. The body weight of subjects was therefore not followed during the intervention study. Incidentally, most of the subjects provided stools of the healthy normal type (types 3 and 4, 55/75 = 73%), and abnormal stool types were randomly distributed among the three clusters, thus stool type was not a major factor for consideration in the clustering of microbiomes in this study.

Singapore is a multiethnic global city. It is not surprising that despite the Asian ethnicity, 41% of the young adults opted for a Western dietary habit, consuming more sugary beverages, meat, and carbohydrates, often in the form of Western fast foods, such as hamburgers, fried chicken, and potato. Their faecal microbiome is denoted as Cluster 1, dominated by *Bacteroides* (30% of total OTUs), having a microbiota profile like meat-eating Europeans and East Asians [11,12,23,24]. However, 19% of the young adults indulged in traditional Southeast Asian dietary habits, consuming more plant-rich foods, such as vegetables and fermented foods. Their faecal microbiome is denoted as Cluster 2, which was dominated by *Prevotella* (21% of total OTUs), typical of Southeast Asians [10,11]. The remaining belonged to Cluster 3 (40% of total subjects studied), who consumed an intermediary frequency of sugary beverages and vegetables in comparison with Clusters 1 and 2. The high consumption frequency of carbohydrate-rich and protein-rich foods is a reflection of regular consumption of both rice and bread (bun) or potato, and meat as burger and slices cooked with vegetables. Cluster 3 subjects did not possess a predominant bacterium but harboured a comparatively higher proportion of *Blautia* (16% vs. Cluster 1 and Cluster 2, 12% each), *Ruminococaceae* (8% vs. Cluster 1, 3% and Cluster 2, 6%) and *Ruminococcus* (4% vs. Cluster 1 and Cluster 2, 1% each) and thus demonstrated a more evenly distributed microbiome profile. In earlier studies, subjects with a microbiome rich in mucin degrading *Ruminococcus* [12,18] were grouped under an enterotype purported to consume more alcohol and polyunsaturated fats.

Between the two predominant gut microbiota, *Prevotella* may have been diet determined, as it showed a strong correlation with most of the food types (carbohydrate, protein, vegetable, nuts and sugary beverages). The *Bacteroides* may be the secondary respondent, in response to the level of *Prevotella*.

The *Lactobacillus* consumed appeared to either reproduce or enhance the reproduction of endogenous *Lactobacillus* strains in the gastrointestinal tract of subjects in Cluster 1 and 3, as there was an increased abundance of *Lactobacillus* in the faecal samples during the consumption period in the *Lactobacillus* intervention study. The study was not able to verify the origin of this increase in *Lactobacillus* as strain-specific primers were not available. The abundance of *Lactobacillus,* however, could not be sustained and returned to that of the basal level within 14 days of stopping oral supplementation. This suggests that the *Lactobacillus* consumed could not colonise the gastrointestinal tract, a sign of colonisation resistance in the clusters.

The *Lactobacillus* was unable to propagate and colonise the gastrointestinal tract of Cluster 2 individuals, as the abundance of *Lactobacillus* in the faecal samples did not alter significantly throughout the *Lactobacillus* administration. This demonstrates that persistence and colonisation of the exogenous *Lactobacillus* in the gastrointestinal tract of Singapore young adults are basal microbiome and diet dependent. This agrees with our earlier studies [25,26].

The introduction of *Lactobacillus* was not able to alter the profile of major microbiota (>1% of total OTUs) in Cluster 1, demonstrating a stable microbiota structure. Consumption of the *Lactobacillus* led to a minimal alteration in the level of cytokines. In short, the *Lactobacillus* showed minimal effects on Cluster 1 subjects in terms of microbiota composition and modulation in immune activities.

The introduction of the *Lactobacillus* to subjects in Cluster 2, however, reduced the abundance of *Prevotella* at Timepoint 4 and rendered the microbiota profile to resemble that of Cluster 3. *Lactobacillus* has been reported inhibitory to *Prevotella* [27,28]. The basal faecal water cytokine IL-2, -12 and TNFα levels in Cluster 2 were 3–6 times higher than the basal levels in Cluster 1. This may imply higher lymphocytic and pro-inflammatory activities in this cluster. This agrees with the proposal that gut *Bacteroides* exclude and protect the host from gastrointestinal pathogens [9], thus it may not be surprising that subjects in Cluster 2 depended on immunity for protection. The immunological activities were modulated upon consumption of the *Lactobacillus*, and the pro-inflammatory activities in the gastrointestinal tract seem to migrate from IL-8 neutrophil-mediated (down-regulated 3.8 times at Timepoint 4) to an IL-1β leukocytic mediated (up-regulated 7.2 times) pathway. The clinical relevance is, however, unclear.

Subjects in Cluster 3 consumed Western and Asian foods in rotation randomly, which is typical among Singaporeans [29]. Thus, a wide variation in the food component profile was observed among the subjects. A Singaporean may consume local Southeast Asian breakfast, English morning tea, American fast food lunch, a Japanese snack for afternoon tea, Chinese style dinner, Indian supper, and the sequence may alter the following day. This alternating dietary habit was not able to support a predominant microbiota, as in Clusters 1 and 2. The vacated ecological niches in the gastrointestinal tract were occupied by small but significant increases in the relative abundance of some major microbiota (>1% total OTUs), namely *Blautia, Ruminococaceae* and *Ruminococcus*.

Gut microbiota diversity has been proposed as beneficial in maintaining the physiological functions and health of the human host [30,31]. Gut microbiota composition is largely determined by diet [10,11,12,13], however, it is difficult to achieve a sufficiently broad range of food types in one single meal to support microbiota diversity. Thus, it is expected that the human gut microbiome is mostly dominated by one microbiota, such as *Bacteroides* or *Prevotella*. The Singapore dietary habit of rotating food types represents an achievable and enjoyable approach in diversifying dietary components and even gut microbiota composition.

The Cluster 3 microbiome structure was not as stable as Cluster 1. Nevertheless, the introduction of the foreign bacterium led only to gradual reduction in the abundance of *Bifidobacterium* and *Collinsella*, but no change among the other major microbiota (>1% total OTUs), and it exhibited colonisation resistance to *Lactobacillus,* as the level of the exogenous *Lactobacillus* was reduced to that of the baseline after cessation of *Lactobacillus* consumption. It should be highlighted that both the alpha and beta diversities (weighted and unweighted) in Cluster 3 across the three time points (during consumption and after cessation of consumption of a large dosage of *Lactobacillus*) remained unchanged, which suggested that the microbiome in Cluster 3 was structurally as stable as that of Clusters 1 and 2. For immunity, except IL-1β at Timepoint 3, consumption of the *Lactobacillus* led to a general reduction of overall lymphocytic and pro-inflammatory activities.

From the perspective of lactic acid bacterium administration, the effect of the *Lactobacillus* on the gut microbiome was basal microbiome dependent. This agrees with our earlier study on the provision of a *Lactobacillus* to people across a wide distance of geographical area, from Mongolia to Singapore [25]. Moreover, the present study showed that immune modulation elicited by the *Lactobacillus* administration was also basal microbiome or immune status dependent. *Lactobacillus* appeared to have a larger immunoregulatory effect on people who belong to Clusters 2 and 3.

The dominating microbiota *Bacteroides* in Cluster 1 has been aligned with health benefits described in the Introduction; however, it is also listed as an independent high-risk factor for many common diseases in developed countries. These include arterial diseases [32,33], type-2 diabetes [34,35], colorectal cancer [36,37,38], cardiomyopathy [39], rheumatoid arthritis [40], inflammatory bowel disease [41], Parkinson’s disease [42], celiac disease [43] and Alzheimer disease [44]. On the other hand, Cluster 2 microbiome is dominated by *Prevotella.* Despite the fact that it positively interferes in energy homeostasis and glucose control [7,8], *Prevotella* is also implicated directly or indirectly in the causation of many chronic inflammatory diseases. These include periodontitis, bacterial vaginosis, rheumatoid arthritis, metabolic disorders [45], tonsillitis [46], advanced fibrosis non-alcoholic fatty liver disease (NAFLD) [47], cardiometabolic risk [48] and asthma [49]. This study thus brings about a critical question. Since the predominating bacterium, *Bacteroides* and *Prevotella* in Clusters 1 and 2 respectively, is associated with some forms of health benefit but also illness, and Cluster 3 microbiome demonstrated a more even distribution of major microbiota types and colonisation resistance, could Cluster 3 be a healthier microbiome structure? This warrants further study, as a means for the maintenance of healthy status. The relatively high economic standing and excellent medical care may explain Singaporeans being among the world’s longest life expectancies at birth [50] (83.5 years according to the 2019 latest data). It could not be ruled out that a diverse microbiome without predominant disease-inducing microbiota contributes to Singaporeans having the status of the longest Healthy Life Expectancy (HALE) at birth [51] (76.2 years according to the 2016 latest data).

As demonstrated in this study, understanding the correlation between the gut microbiome, diet and health in multiethnic communities represents an excellent model system, due to the wider variety of food choices in a population of defined ethnicity, lifestyle and living environment. In summary, alternating dietary types over meals, as in Singaporean dietary habits, represents a practical approach in achieving a broad base dietary composition. This appeared to balance the gut microbiota profile, leading to a non-predominant microbiota type (enterotype).

## Figures and Tables

**Figure 1 microorganisms-09-00702-f001:**
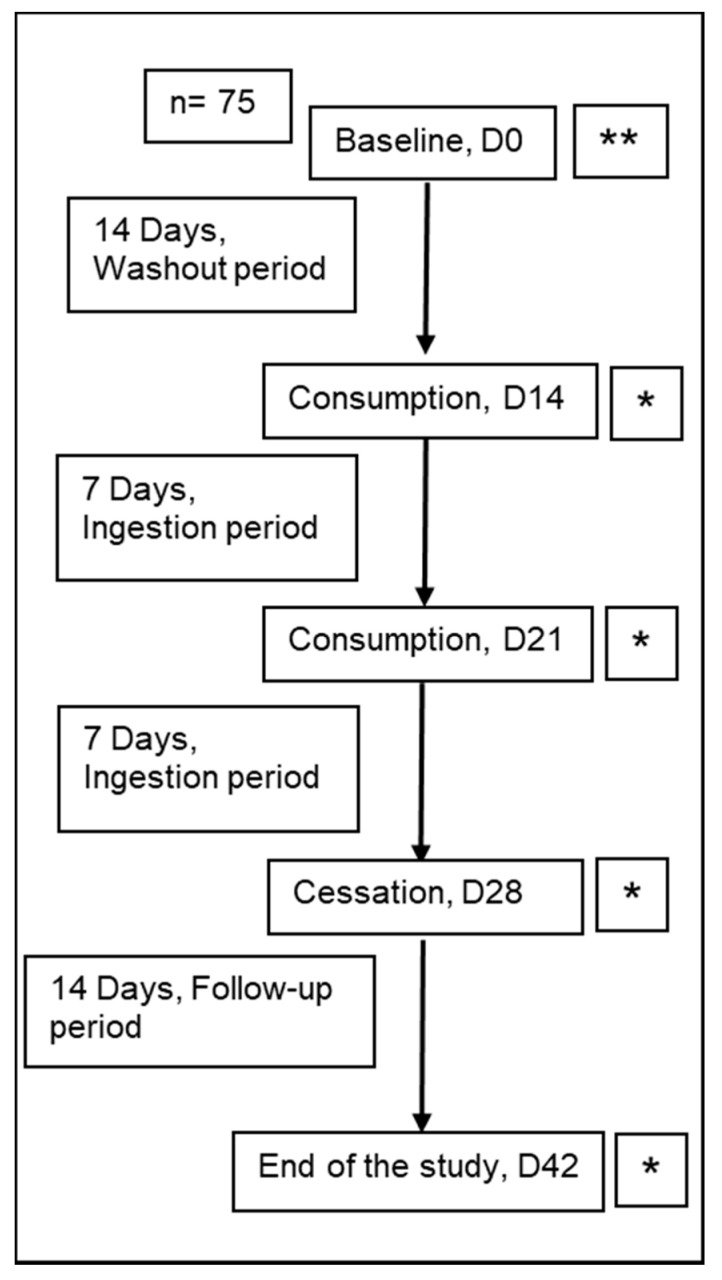
Flow chart of study design. Total 42 days of study, 14 days washout period, 14 days ingestion period and 14 days follow-up period. ** Food Frequency Questionnaire. * Faecal samples collection. D = Day, *n* = 75.

**Figure 2 microorganisms-09-00702-f002:**
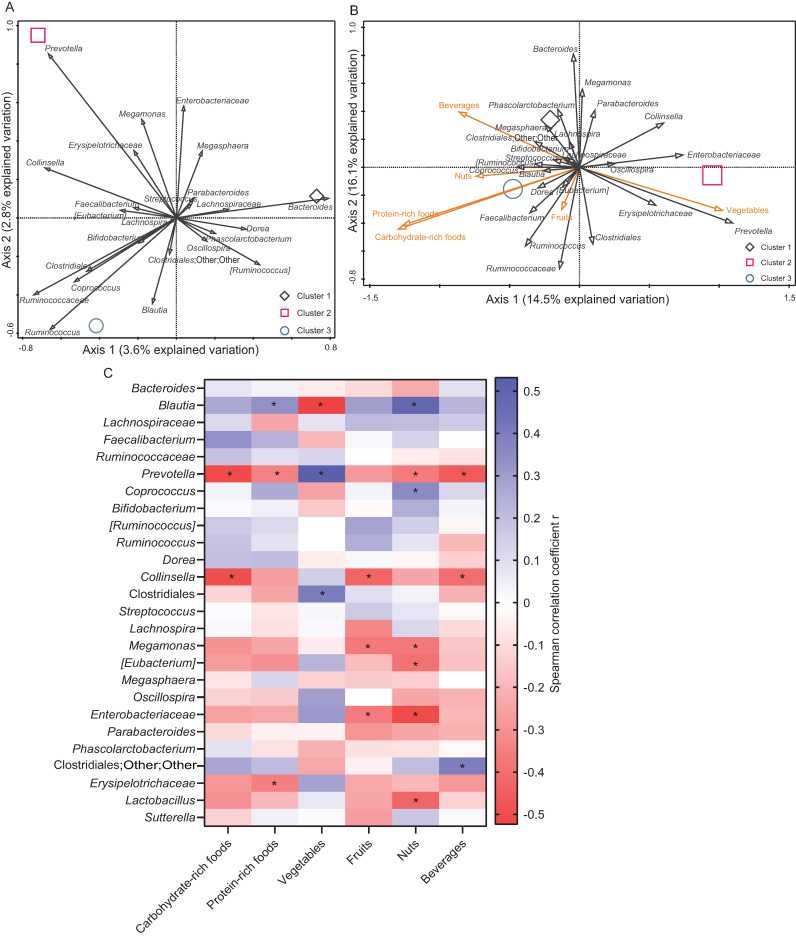
(**A**) Composition of major faecal microbiota and (**B**) distribution of dietary consumption and major faecal microbiota in three clusters at baseline. Species biplots of distance-based redundancy analysis (db-RDA) based on the square root Bray-Curtis distance matrix showed the distribution of faecal bacterial genera and the frequency of food items consumption per week in three clusters. The relative abundance of major (>1%) faecal bacterial genera (black arrows) and dietary consumption (orange arrows) shown here are concentrated with the respective arrow. The percentages of axes explain the compositional variation of the respective axis. The distances between each pair of clusters were tested for significance by permutational multivariate analysis of variance (PERMANOVA) and Bonferroni’s multiple comparison pairwise test at 4999 permutations. Different symbols represent the types of clusters. (**C**) Correlation between the frequency of dietary consumption per week (*X*-axis) and major (>1% of total OTUs) faecal bacteria (*Y*-axis) at baseline. Spearman correlation coefficient r values in gradient scales were plotted and presented as a heatmap. The significantly different correlations (two-tailed *p* < 0.05) were marked with asterisks *.

**Figure 3 microorganisms-09-00702-f003:**
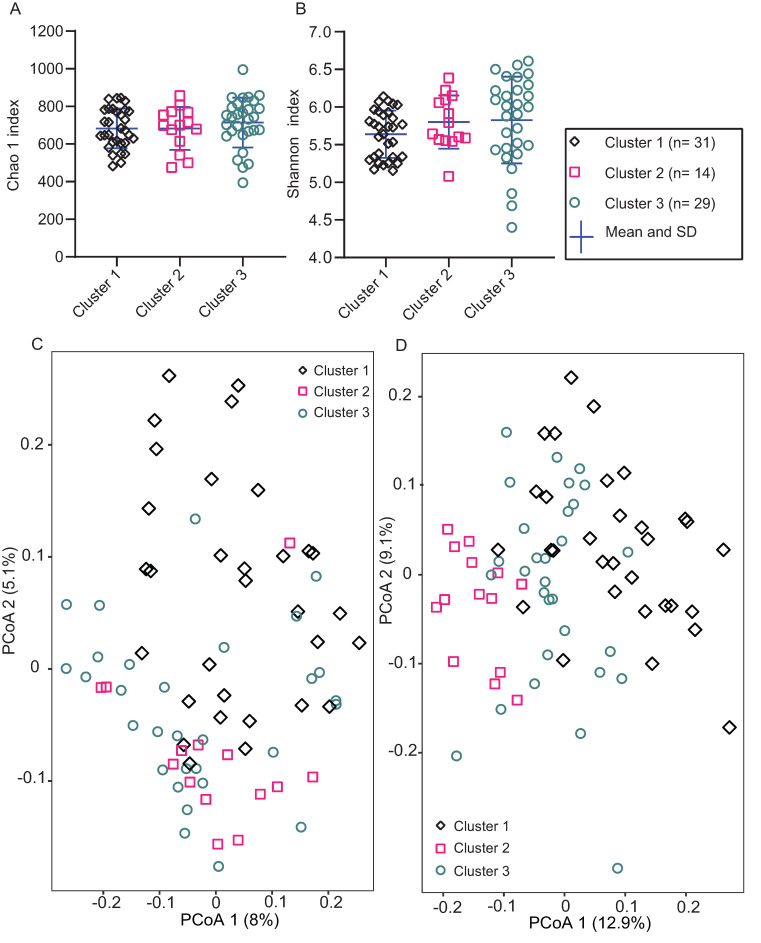
(**A**) Chao 1′s and (**B**) Shannon’s indices of alpha diversity comparing three clusters at baseline. (**C**) Unweighted, (**D**) weighted Unifrac principal coordinates analysis (PCoA) for beta diversity comparing three clusters at baseline. Different symbols and colours represent different clusters. Means and SD of indices are presented. One-way analysis of variance (ANOVA) followed by Bonferroni multiple comparisons tests were applied and the indices were not significantly different between the clusters at baseline of each cluster. The distances between each cluster were tested for significant difference by permutational multivariate analysis of variance (PERMANOVA) and Bonferroni’s multiple comparison pairwise test at 4999 permutations. Timepoint 1 = baseline; 14 days after washout, SD = standard deviation. Numbers of samples in parenthesis.

**Figure 4 microorganisms-09-00702-f004:**
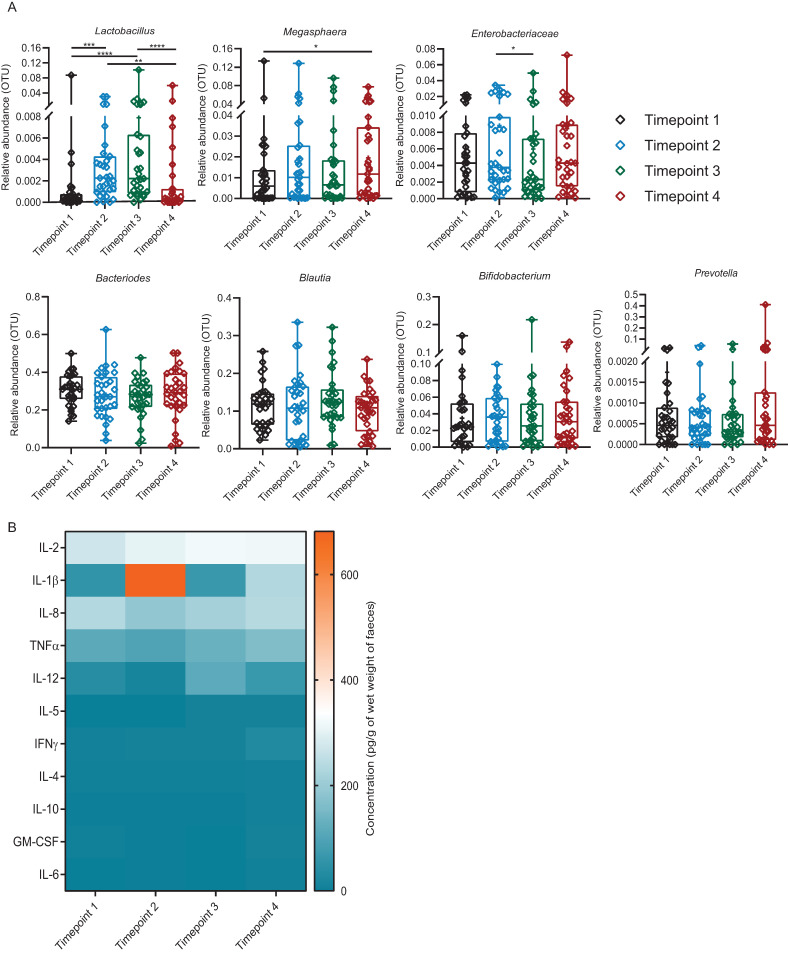
Profiles of faecal bacterial genera and cytokines of Cluster 1. (**A**) Comparison of relative abundance (OTUs) of seven major faecal bacterial genera of Cluster 1 across the time points. Different colours represent different time points. The bacteria which were significantly different between each time point (tested by Friedman rank-sum and post hoc Nemenyi multiple pairwise comparison tests) are presented as **** *p* < 0.0001, *** *p* ≥ 0.0001–*p* < 0.001, ** *p* ≥ 0.001–*p* < 0.01, * *p* ≥ 0.01–0.05. (**B**) Comparison of concentration of faecal water cytokines (pg/g of wet weight of faeces) of Cluster 1 across the time points. Means of concentration are presented. No significant difference in the cytokine levels between the time points were found by two-way analysis of variance (ANOVA) followed by Bonferroni multiple comparisons tests. IL = Interleukin, TNF = Tumour necrosis factor, IFN = Interferon, GM-CSF = Granulocyte-macrophage-stimulating factor. Timepoint 1 = baseline; 14 days after washout, Timepoint 2 = first 7 days after ingestion, Timepoint 3 = second 7 days after ingestion, Timepoint 4 = follow-up; 14 days after non-ingestion. *n* = 31 for each time point.

**Figure 5 microorganisms-09-00702-f005:**
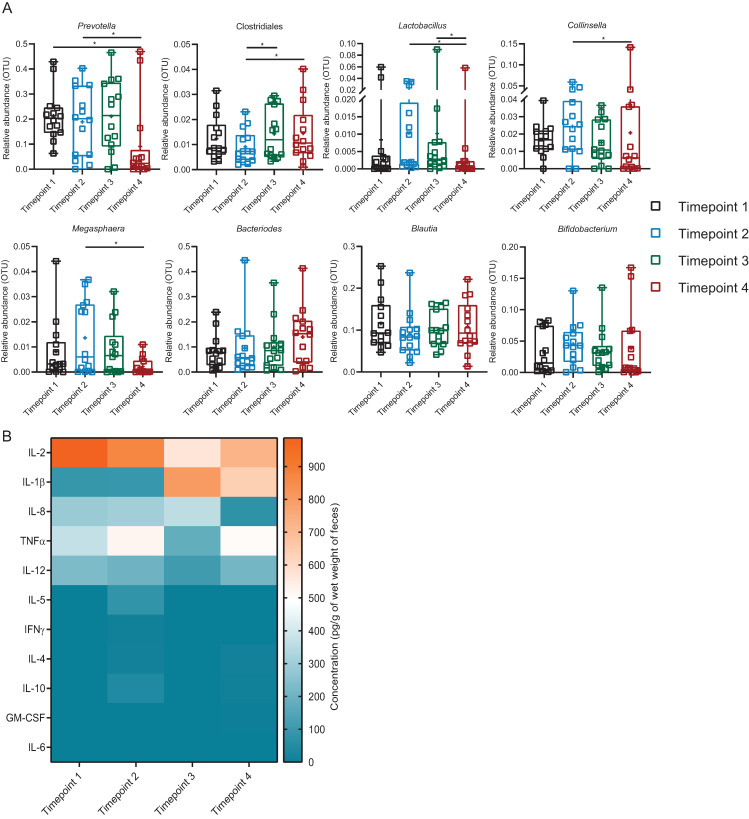
Profiles of faecal bacterial genera and cytokines of Cluster 2. (**A**) Comparison of relative abundance (OTUs) of eight major faecal bacterial genera of Cluster 2 across the time points. Different colours represent different time points. The bacteria which were significantly different between each time point (tested by Friedman rank-sum and post hoc Nemenyi multiple pairwise comparison tests) are presented as * *p* ≥ 0.01–0.05. (**B**) Comparison of concentration of faecal water cytokines (pg/g of wet weight of faeces) of Cluster 2 across the time points. Means of concentration are presented. No significant difference in the cytokine levels between the time points were found by the two-way analysis of variance (ANOVA) followed by Bonferroni multiple comparisons tests. IL = Interleukin, TNF = Tumour necrosis factor, IFN = Interferon, GM-CSF = Granulocyte-macrophage-stimulating factor. Timepoint 1 = baseline; 14 days after washout, Timepoint 2 = first 7 days after ingestion, Timepoint 3 = second 7 days after ingestion, Timepoint 4 = follow-up; 14 days after non-ingestion. *n* = 14 for each time point.

**Figure 6 microorganisms-09-00702-f006:**
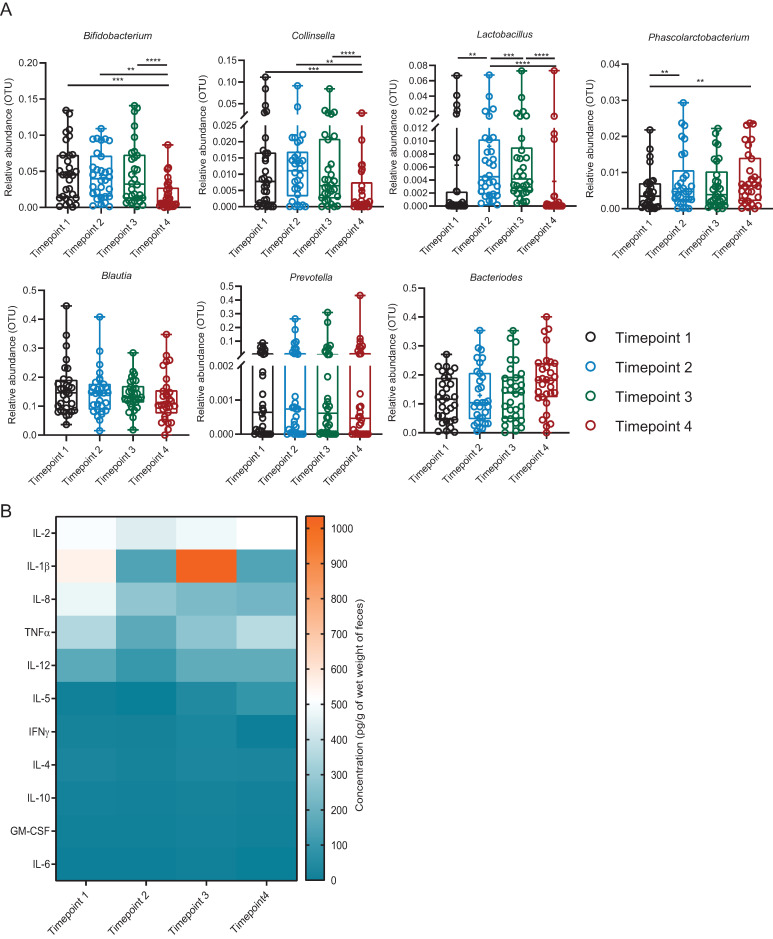
Profiles of faecal bacterial genera and cytokines of Cluster 3. (**A**) Comparison of relative abundance (OTUs) of seven major faecal bacterial genera of Cluster 3 across the time points. Different colours represent different time points. The bacteria which were significantly different between each time point (tested by Friedman rank-sum and post hoc Nemenyi multiple pairwise comparison tests) are presented as **** *p* < 0.0001, *** *p* ≥ 0.0001–*p* < 0.001, ** *p* ≥ 0.001–*p* < 0.01. (**B**) Comparison of concentration of faecal water cytokines (pg/g of wet weight of faeces) of Cluster 3 across the time points. Means of concentration are presented. No significant difference in the cytokine levels between the time points were found by the two-way analysis of variance (ANOVA) followed by Bonferroni multiple comparisons tests. IL = Interleukin, TNF = Tumour necrosis factor, IFN = Interferon, GM-CSF = Granulocyte-macrophage-stimulating factor. Timepoint 1 = baseline; 14 days after washout, Timepoint 2 = first 7 days after ingestion, Timepoint 3 = second 7 days after ingestion, Timepoint 4 = follow-up; 14 days after non-ingestion. *n* = 30 for each time point.

## Data Availability

All data relevant to this study were submitted as supplementary tables and the sequencing data were deposited at EBI repository (accession no: PRJEB39242).

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
