# Peer review of "Gut Microbiome of a Multiethnic Community Possessed No Predominant Microbiota"

_microorganisms, 2021, doi:10.3390/microorganisms9040702_

Round 1
Reviewer 1 Report
Abstract: Line #14- in the abstract needs to be rephrased. Overall the abstract sounds confusing. Please write shorter sentences and discuss the background in more to cause impact. Introduction There are a few grammatically incorrect sentences in the introduction, please rectify them. Also, the introduction should be expanded a little more to reflect the background of the study, significance of the study, breakthroughs in the field so far and concluding in specific issues that the authors have addressed in the manuscript. The implications of a gut dominance of the species mentioned needs to be elaborated upon. Line # 40 sounds controversial as easy to misinterpret. Please rephrase it. Other parts of the manuscript has certain word choice and grammatical errors. if the diversity in the discussion can be expanded the study can be correlated with a larger population. The first few points of the discussion would suit limitations better. Also, if there is any information regarding probiotic consumption, that could potentially substantiate or sway the findings.Author Response
Please see the attachment.

Reviewer 2 Report
The authors Khine et al. have investigated where globalization has any effect on the gut microbiome/cytokine levels in a multi-cultural study w/, w/o probiotic Lactobacillus casei for 14 days. They found 3 distinct clusters with different microbial compositions based on diets without any predominant bacteria.
Comments
- The authors have included both the sexes in the work, have the authors found any sex-based difference in microbiome or cytokine levels?
- Why did the authors decide to use Lactobacillus casei, what the strain ID?
- Dosing, how did the authors decide 14-day and 10^8 cfu dose? The probiotics are generally dosed per body weights?
- Did the authors find any difference in the body weights and BMI w/, w/o probiotics? Perhaps it is best look at any associations? There was no detailed mention.
- In the figure 2, Nuts where significantly correlated with genus lactobacillus. What are the nuts involved and was the change more with nuts?
- What are the inclusion and exclusion criteria? Did the authors look for the participants diet charts whether or not if they have been consuming fibers, fermented foods or antibiotic use ?
- What are the limitations of the study? Where the participants recruited in a single site or multiple?
Other comments:
I suggest the authors provide a study design illustration. The terms timepoints and clusters, although briefly mentioned in the legends led to a state of confusion. Figure 4 is highly difficult to follow with clusters. Also, please enhance the quality of figures for proper viewing.
The word "cultural" is vague. Although, the authors used dietary habits along with multicultural substantiation. can be replaced with ethnic society or so, should be double-checked before publication.
Round 2
Reviewer 2 Report
The manuscript is now acceptable. No further comments.